# Chronic Pain in Dogs and Cats: Is There Place for Dietary Intervention with Micro-Palmitoylethanolamide?

**DOI:** 10.3390/ani11040952

**Published:** 2021-03-29

**Authors:** Giorgia della Rocca, Davide Gamba

**Affiliations:** 1Department of Veterinary Medicine, Centro di Ricerca sul Dolore Animale (CeRiDA), Università degli Studi di Perugia, 06123 Perugia, Italy; 2Operational Unit of Anesthesia, Centro Veterinario Gregorio VII, 00165 Roma, Italy; davidegamba@mac.com; 3Freelance, DG Vet Pain Therapy, 24124 Bergamo, Italy

**Keywords:** N-acylethanolamines, palmitoylethanolamide, chronic pain, small animals, micronization, endocannabinoids, microglia, mast cells

## Abstract

**Simple Summary:**

Chronic pain is being increasingly recognized and addressed in small animal practice. The recent recognition that inability to communicate does not negate the possibility to experience pain requires veterinarians to actively recognize, assess and manage animal pain. In order to successfully treat pain while limiting side effects, a combination of different therapeutic weapons (e.g., analgesic drugs, acupuncture, physiotherapy and dietary interventions) is currently preferred. In this perspective, the endocannabinoid-like palmitoylethanolamide represents a promising option, since it is naturally occurring in food sources and animal tissues, addresses the mechanisms of chronic pain (i.e., immune cell hyperactivity) and is presently used in complementary feeds for dogs and cats in highly absorbable micronized formulations (i.e., micro-palmitoylethanolamide). In the present paper, the role of immune non-neuronal cells in chronic pain is reviewed. Moreover, the function of body-own palmitoylethanolamide in controlling pain through non-neuronal cell modulation is discussed. Finally, data on pain-relieving effects provided by dietary supplementation with micro-palmitoylethanolamide are presented. The critical mass of data here reviewed might help veterinary practitioners in the process of evidence-based decision-making regarding the management of chronic pain in cats and dogs.

**Abstract:**

The management of chronic pain is an integral challenge of small animal veterinary practitioners. Multiple pharmacological agents are usually employed to treat maladaptive pain including opiates, non-steroidal anti-inflammatory drugs, anticonvulsants, antidepressants, and others. In order to limit adverse effects and tolerance development, they are often combined with non-pharmacologic measures such as acupuncture and dietary interventions. Accumulating evidence suggests that non-neuronal cells such as mast cells and microglia play active roles in the pathogenesis of maladaptive pain. Accordingly, these cells are currently viewed as potential new targets for managing chronic pain. Palmitoylethanolamide is an endocannabinoid-like compound found in several food sources and considered a body’s own analgesic. The receptor-dependent control of non-neuronal cells mediates the pain-relieving effect of palmitoylethanolamide. Accumulating evidence shows the anti-hyperalgesic effect of supplemented palmitoylethanolamide, especially in the micronized and co-micronized formulations (i.e., micro-palmitoylethanolamide), which allow for higher bioavailability. In the present paper, the role of non-neuronal cells in pain signaling is discussed and a large number of studies on the effect of palmitoylethanolamide in inflammatory and neuropathic chronic pain are reviewed. Overall, available evidence suggests that there is place for micro-palmitoylethanolamide in the dietary management of chronic pain in dogs and cats.

## 1. Introduction

The revised definition of pain endorsed and approved by the International Association for the Study of Pain (IASP) defines pain as “An unpleasant sensory and emotional experience associated with, or resembling that associated with, actual or potential tissue damage” [1]. An important change with respect to the previous definition (1979) consists in the recognition that verbally expressing pain is no more a prerequisite to experiencing pain. The IASP further explains that “Verbal description is only one of several behaviors to express pain; inability to communicate does not negate the possibility that a human or a nonhuman animal experiences pain” [1].

From a veterinary perspective, this represents a definitive recognition of animal pain and poses veterinary practitioners in an “algological position”, i.e., to play a proactive role in recognizing, assessing and managing animal pain. Indeed, many efforts have been made in this direction during the last decades and several European and US groups are moving toward the development of better protocols to detect [2,3,4,5,6,7], measure [8,9,10,11,12,13,14] and treat [7,15,16,17,18] animal pain accordingly. The ever-increasing availability of well-designed pain scales for acute and chronic pain in dogs and cats [19,20,21,22,23,24,25,26] and the Pain Management Guidelines [27,28] are good examples.

On the treatment side, one of the most up-to-date and clinically relevant issues consists in the multimodal approach to pain management, i.e., a combination of different therapeutic weapons, like analgesic drugs, acupuncture and physiotherapy techniques, as well as dietary interventions [29,30,31,32,33]. With regard to the last measure, calorie restriction and omega-3 fatty acids are the most investigated approaches to chronic pain in pets, particularly osteoarthritis pain [34,35].

Increasing evidence is accumulating on the beneficial effects of N-acylethanolamines (NAEs) in chronic pain. NAEs have been detected in several food sources of vegetable [36,37,38] and animal origin [39]. Moreover, chronic or subchronic high-fat diet, as well as deficient intake of essential fatty acids have been shown to profoundly affect NAE levels in animal body [40,41,42,43,44,45]. One of the most studied NAEs is the endocannabinoid-like mediator palmitoylethanolamide (PEA). Its levels in food sources and its pro-homeostatic role have been recently reviewed [46].

Interestingly, the autoprotective function of PEA was first suggested in dogs, when it was found that the canine myocardium produced PEA in response to ischemic injury [47] and canine brain possessed the biosynthetic and degradative machinery for PEA [48,49]. Since then, an increasing body of literature has emerged highlighting the importance of dietary intervention with micro-PEA—i.e., the bioavailable form of PEA—for pain relief [50,51,52,53].

The present paper outlines current information on the involvement of immune cells in chronic pain and reviews the role of endogenous PEA in pain control, as well as the experimental and clinical data on pain relieving effects provided by different PEA formulations.

Given that some micro-PEA-containing dietary supplements for dogs and cats are currently being available on the European market, this review wishes to provide scientific evidence to make informed decisions about the management of chronic pain in cats and dogs.

## 2. Pain Classification

Pain includes at least two dimensions, i.e., physical and emotional components. From a physical perspective, although pain is often conceived as a homogeneous sensory entity, several distinct types exist: transient, inflammatory, neuropathic and functional pain (Figure 1) [54].

Transient pain develops when a potentially harmful insult is applied to a superficial or deep tissue (cutaneous/mucous or musculoskeletal/visceral, respectively) for such a short time that it does not cause tissue damage (potential damage). It develops rapidly and has a transient nature, disappearing with the end of the harmful stimulus or shortly thereafter. Transient pain acts like an alarm signal, capable of activating a sudden withdrawal reflex that protects the tissues from the noxious stimulus (adaptive pain). It develops thanks to the activation of the nociceptive system and the transduction, transmission, modulation and integration events that follow (nociceptive pain).

Inflammatory pain derives from damage-induced inflammation to somatic or visceral tissues (actual damage). It can be acute or chronic, depending on the nature of the underlying disease. While the former still has a protective purpose, as it limits movements and further damage until the repair is completed (adaptive pain), the latter lacks any biological purpose (maladaptive pain). Inflammatory pain is the result of nociceptor activation by inflammatory soup mediators released from immune cells, mainly mast cells. This leads to the development of neurogenic inflammation and brings about subsequent neurochemical changes, like wind-up and long-term potentiation, as well as translational and transcriptional modifications (e.g., lower activation threshold of nociceptors and increased expression of functional proteins involved in pain processing). The increased firing rate of the first and projection neuron (i.e., peripheral and central sensitization, respectively) is the main feature of inflammatory pain and leads to primary or secondary hyperalgesia (i.e., increased response to painful stimuli at the site of, or distant to the stimulus) and allodynia (i.e., painful response to harmless stimuli).

Neuropathic pain is defined by IASP as “pain that arises as a direct consequence of a lesion or disease affecting the somatosensitive system”. It results from an abnormal activation of the pain pathways, due to a dysfunction or damage to peripheral nerves and/or dorsal nerve roots (peripheral neuropathic pain) as well as spinal cord and/or brain (central neuropathic pain). Accordingly, it is considered disnociceptive, acquires a pathological, maladaptive nature and can be viewed as a disease itself rather than a symptom. Neuropathic pain can last for months to years or possibly even a lifetime, being thus considered a type of chronic pain. Possible mechanisms of peripheral neuropathic pain are (i) persistent hyperexcitability of nociceptors (even after damage repair), (ii) increased excitability of nociceptive fibers following nerve damage (e.g., after dysmyelinosis or neuroma formation), and (iii) structural/functional changes of spinal synapses following nerve degeneration. The resulting burst stimulation of afferent fibers may lead to central sensitization, a hallmark of several painful disorders like feline osteoarthritis [55]. Central neuropathic pain involves spinal cord and supramedullary neuronal structures and results from lesions affecting the central nervous system or increased activity of thalamic and cortical neurons due to neurochemical changes (e.g., imbalance of glutamatergic/GABAergic transmission).

Functional pain occurs spontaneously, in the total absence of tissue damage or evident dysfunction or damage to the nociceptive nervous system. It is probably supported by persistent plastic modifications of the central neuronal circuits induced by nociceptive or dysnociceptive algogenic lesions. As a consequence, originally activated central neuronal circuits remain active even when the lesion has resolved. A possible hypothesis is that mechanisms underlying the spontaneous processing of pain are similar to those that underlie memory: modifications of central neuronal circuits, initially induced by tissue or nerve damage, would remain in the CNS as traces of memory and can be “remembered” even after the lesion has resolved. Functional pain is therefore non-nociceptive, it can last months, years or forever, establishing its chronic nature. It has no biological function and is rather pathological (maladaptive). Like neuropathic pain, functional pain can thus be viewed as a disease itself [29,54,56].

## 3. Role of Non-Neuronal Cells in the Development and Resolution of Chronic Pain

As introduced above, chronic pain is an unpleasant experience outlasting the time of healing. Particular cells of the immune system intimately associated with or located within the nervous system, i.e., “non-neuronal cells”, are increasingly acknowledged as major contributors to the development and maintenance of chronic pain [51]. In particular, mast cells (within the nervous system and in the periphery) and microglia (at spinal and supraspinal level) interact with neurons under physiological and pathological conditions (Table 1).

While in the first situation non-neuronal cells support the well-function of neurons (e.g., through releasing neurotrophic factors), in the latter they may become hyper-activated and may cause pain to continue after the original injury has healed [94,95]. In fact, prolonged activation of non-neuronal cells leads to uncontrolled release of pro-inflammatory mediators resulting in long-lasting plastic changes of synaptic connectivity, with enhanced transmission of nociceptive information, alterations of pain signaling pathways and chronic pain development [96,97].

It should also be considered that a bidirectional crosstalk between mast cells and microglia exists [98] and is currently acknowledged as a critical event in pain hypersensitivity [64,99]. Accordingly, non-neuronal cell hyper-activation—and the resulting neuroinflammation—is a key player of pain states (Figure 2) [100,101,102].

Interestingly, non-neuronal cells are also endowed with crucial protective functions in resolution of neuroinflammation and pain [59]. Indeed, mast cells and microglia are able to reduce sensitization by producing pro-resolution mediators, the so-called specialized pro-resolving lipid mediators [103,104,105].

In this framework, particular attention is currently devoted to endocannabinoids and related lipid compounds, such as NAEs and more particularly PEA [106,107,108,109]. As detailed below, PEA and similar endocannabinoids are locally released on demand during injury to counterbalance the effects of pro-algesic mediators [110,111].

## 4. Endogenous PEA and Pain Modulation

As briefly introduced above, non-neuronal cells not only dangerously boost pain signaling, but also exert crucial functions in resolution of neuroinflammation and pain, through pro-resolution mediators. Among them, endocannabinoids and related NAEs are increasingly being acknowledged to play key roles in pain modulation, with PEA being one of the most studied [112]. It has been repeatedly found in dozens of vegetable and animal food sources (in nanogram per gram level), from soy to carrots and from eggs to beef [39,46]. Moreover, PEA levels have also been detected in virtually any tissue and body fluid [46,51], where it is enzymatically produced “on demand” in response to actual or potential damage and enzymatically cleaved when it has served its purpose [51,52,113,114,115].

The late Nobel prize winner Rita Levi Montalcini first proposed that PEA acts as an Autacoid Local Injury Antagonist (ALIA), through down-modulating mast cell degranulation [116,117]. It was then found that PEA is synthesized by mast cells and microglia [118,119] and is able to keep cell reactivity within physiological boundaries [51], thereby controlling neuroinflammation and chronic pain [120,121,122].

It has also been demonstrated that PEA not only acts through non-neuronal cells, but may also directly influence neurons. Indeed, PEA was shown to (i) exert protective effects on cultured cortical and cerebellar neurons [123,124], (ii) control spontaneous GABAergic synaptic activity in striatal neurons [125], (iii) dose-dependently increase intracellular calcium concentration in sensory neurons thereby desensitizing pain receptors [126]; (iv) modulate the activity of dorsal root ganglion neurons [127].

On the molecular side, PEA controls neuronal and non-neuronal cells through direct or indirect receptor targets, ranging from canonical to putative cannabinoid receptors, i.e., cannabinoid receptor type 1 and 2 (CB1 and CB2), peroxisome proliferator-activated receptor α (PPARα), transient receptor potential vanilloid 1 (TRPV1) and G protein-coupled receptors 55 and 119 (GPR55, GPR119) [65,126,127,128,129,130]. The indirect receptor agonism—i.e., a particular kind of entourage effect—depends on PEA ability to increase the local levels of the endocannabinoids anandamide (AEA) and/or 2-arachydonoylglycerol (2-AG) [52,125,131,132]. Different types of cannabinoid receptors have been recently localized in canine and feline central and peripheral organs. In particular, the target receptors of PEA have been found in canine and feline skin [133,134,135], along the gastrointestinal tract [136,137,138], in different brain areas [139,140,141,142], spinal cord and dorsal root ganglia [141,143]. The distribution of cannabinoid receptors in dogs and cats has been recently addressed by Gugliandolo et al. [46], to whose paper the reader is referred for more detailed information.

The multiple receptor mechanism(s) of PEA is responsible for innate pain control (Figure 3) [46,52] and provides PEA with a natural analgesic function, originally proposed in the late 1990s by Calignano and colleagues [144] and later even better designed by Piomelli and Sasso [145].

Currently, the role of body-own PEA in pain control is unquestionably proven by the recent case of a pain-insensitive woman who lacks the NAE degradative enzyme [146]. She feels almost no pain and has much higher levels of NAEs, with PEA levels being around 4-fold higher than normal.

In summary, PEA is an endogenous compound endowed with pain-relieving functions. It is locally produced on demand by non-neuronal cells and other cell types in response to an actual or potential damage, and acts as an endocannabinoid direct or indirect agonist to keep non-neuronal cell response within homeostatic boundaries.

## 5. Causes and Prevalence of Maladaptive Pain in Dogs and Cats

In the last decades, pets are becoming an increasingly important part of family life, being often considered real family members. Owners are more and more often seeking veterinary attention for various diseases affecting their pets, including pain. However, while most information on pain control in dogs and cats exists regarding peri-operative analgesic use, chronic pain conditions are still being undiagnosed and under-treated, especially in the feline species [147].

Indeed, many conditions may cause maladaptive pain in dogs and cats, as summarized in Table 2.

The incidence of pain in dogs and cats has not received much attention so far. A cross-sectional study on 317 dogs and 112 cats admitted to an emergency service reported that 56% and 54% of dogs and cats respectively were painful, with most dogs suffering from deep somatic pain and most cats from visceral pain [149]. The percentage was lower in outpatients (1153 dogs and 652 cats), with 20% of dogs and 14% of cats showing evidence of pain [150]. Neuropathic pain was diagnosed in 7–8% of both species [150,151].

Among the causes listed in Table 2, one of the most frequent painful conditions in dogs and cats is osteoarthritis (OA), otherwise referred to as osteoarthrosis or degenerative joint disease. The prevalence of canine OA published so far varies widely. In the UK, estimates range from 6.6% in primary-care services [152,153] to 20% based on referral data [154]. Estimates from North America made on radiographic and clinical data from referral settings show the age-specific prevalence of canine OA, with values ranging from 20% in dogs older than one year to 80% in dogs over eight years [155]. A cross-sectional study on radiographic signs of feline OA showed an overall prevalence of 92% in randomly selected domestic cats (mean age of 9.9 years) [156,157].

Finally, it should be mentioned that recognizing and measuring pain in animals is anything but easy. Further complicating the issue is the discovery that people rate pain sensitivity differently based on breed-specific stereotypes or phenotypic traits and dog breed archetypes [158]. Many excellent review papers are available on pain assessment in companion animals, which the reader is referred to [22,23,24,159,160].

## 6. Management of Pain in Dogs and Cats

As previously discussed, chronic pain—regardless of the underlying cause—may become maladaptive, i.e., without any beneficial role. Neuropathic pain, functional pain and chronic inflammatory pain are all types of maladaptive pain. Any type of maladaptive pain is thus considered pathological and must be treated accordingly.

A full discussion on pain management in pets is behind the scope of this article. Briefly, non-steroidal anti-inflammatory drugs (NSAIDs), opioids and steroids alone or associated with adjuvant drugs such as gabapentinoids (gabapentin, pregabalin), NMDA-antagonists (amantadine, memantine), selective serotonin reuptake inhibitors (SSRIs), serotonin-norepinephrine reuptake inhibitors (SNRIs) or tricyclic antidepressants (TCA, e.g., amitriptyline), among others, represent the mainstream pharmacologic treatment of pain [27,161,162]. However, when used alone or even in combination, these drugs may still fail to provide complete pain relief [149]. Moreover, they can lead to the occurrence of adverse effects, especially in the chronic use [163,164]. Chronic pain in pets thus still represents an unmet medical need.

The idea that multimodal analgesia tailored to the patient will have most chances of being effective is increasingly being acknowledged in veterinary practice [165,166,167]. In this view, dietary intervention with pro-resolving lipid compounds may represent an ideal adjunctive approach. PEA is currently one of the most promising options in this regard.

## 7. PEA and Formulation Challenges: A Size Issue

Before dealing with the effectiveness of PEA in chronic pain, a key formulation question must be addressed. PEA is a highly lipophilic compound and tends to aggregate in large particles (up to 2000 microns)—a big pharmaceutical issue since absorption rate is inversely related to particle size [46,108,168].

Particle size reduction through micronization techniques (down to 0.8 microns) importantly improves the dissolution and thus bioavailability (Figure 4A) [169]. This results in superior efficacy of orally administered PEA (Figure 4B) [170,171,172], while ensuring its safety [171]. Mainly for this reason, in clinical practice (in which oral route is preferred because of ease of administration) the micronized (PEA-m) and ultra-micronized (PEA-um) forms (collectively known as micro-PEA [158]) are privileged and are indeed the most investigated.

On the contrary, in laboratory animals, the intraperitoneal delivery is generally the easiest and most used administration route. Moreover, it results in faster and more complete absorption compared to oral route [173]. This is especially true if suspension in carboxymethyl cellulose is used [173], as it is usually the case with intraperitoneally administered PEA. Indeed, no difference was observed between PEA-um and naïve PEA in pain control, in the event of intraperitoneal delivery (Figure 4C), which is absolutely not the case if oral administration is used [172].

## 8. Preclinical Evidence for PEA in Pain Relief

The rationale to administer PEA for pain relief and wellbeing was brilliantly foreseen in the late nineties by the Nobel Prize Winner Rita Levi Montalcini, who stated that “the observed effects of Palmitoylethanolamide appear to reflect the consequences of supplying the tissue with a sufficient quantity of its physiological regulator of cellular homeostasis” [117].

Since then, several studies in preclinical pain models have been performed, with PEA being given mainly via intraperitoneal route, although intraplantar injection [144,174] and oral administration of micronised formulations [175] were also used. Interestingly, the concurrent administration of micro-PEA and morphine for 11 days attenuated the development of opioid tolerance [176], since micro-PEA strengthens morphine analgesia and allows prolonged and effective pain relief with low doses [177].

Moreover, a descending analgesic mechanism mediated by the serotonergic system has been suggested [178].

Table 3 and Table 4 summarize the main preclinical investigations. As shown, PEA exerts a clear anti-nociceptive effect in chronic pain models of either inflammatory [132,144,174,175,176,179,180,181,182,183,184,185,186,187,188,189,190,191], neuropathic [192,193,194,195,196,197,198] or mixed nature [199,200]. In particular, it has been found that the anti-nociceptive effect of PEA is comparable to synthetic or plant-derived cannabinoids used for chronic pain, like nabilone [181] and delta-9-tetrahydrocannabinol (Δ9-THC) (Figure 5) [190].

Interestingly, an in-press study by Tagne and collaborators has just shown that hemp oil extract with 9.3% cannabidiol by weight has little or no effect when administered alone but synergizes with PEA to produce a greater-than-additive alleviation of neuropathic pain, upon single-dose administration (Figure 6) [201]. According to the authors, a possible explanation for the observed synergistic interaction lies in the ability of hemp oil extract to improve pharmacokinetic profile of PEA [201].

Two particularly interesting issues arise from the above findings. First, PEA is not analgesic *sensu stricto* since it does not modify the physiological pain threshold of control animals and rather electively normalizes conditions of hypersensitivity [176,185]. Second, PEA not only relieves pain itself but also improves pain-induced cognitive impairments [198].

As far as mechanism of action are concerned, reduced mast cell hyperplasia—even in endoneural sites—and decreased spinal microglia activation were the main observed events [120,185]. At the molecular level, the reduction of markers of pain pathway activation (e.g., Fos) and inflammatory mediators (e.g., cytokines, nerve growth factor) as well as modulation of extracellular signal-regulated kinase (ERK) and nuclear pro-inflammatory factors (e.g., NF-kB) were detected in the spinal cord [120,132,182,193,195,197]. Restoration of the glutamatergic synapses homeostasis in the prefrontal cortex and the involvement of de novo neurosteroid synthesis (i.e., allopregnanolone) in the spinal cord were also suggested to mediate PEA-induced analgesia [198,202]. Moreover, electrophysiological signs of decreased neuronal hyper-excitability were reported at the spinal cord level of PEA-um treated neuropathic animals [176,185,200]. Finally, the involvement of cannabinoid receptor(s) (e.g., CB2, CB1, PPARα) in the pain-relieving effect of PEA was repeatedly confirmed [186,192,194,199,202,203]. The up-regulation of CB2 expression by microglia through PPARα activation has also been suggested as a possible mechanism underlying the pain-relieving effect of PEA [204].

According to an impressive meta-analysis by IASP Presidential Taskforce on Cannabis and Cannabinoid Analgesia, PPARα agonists and, more specifically, PEA, are effective in attenuating pain-associated behaviors in a broad range of inflammatory or neuropathic pain models [205].

**Table 3 animals-11-00952-t003:** Pain relieving effect of PEA—mainly given via intraperitoneal route—in animal models of chronic inflammatory pain. Summary of studies in chronological order.

Animal Model	Main Behavioural Effect	Ref.
**Somatic Inflammatory Pain**		
Carrageenan-induced hyperalgesia	Significant reduction of mechanical hyperalgesia	[179]
Formalin-induced persistent somatic pain	Significant inhibition of both early and late phases of formalin-evoked pain behaviour	[144]
Formalin-induced persistent somatic pain	Significant reduction of the second phase behavioural response (composite pain score)	[180]
Formalin-induced persistent somatic pain	Marked inhibition of pain behaviour	[174]
Carrageenan-induced hyperalgesia	Abolishment of hyperalgesic response	[181]
Intraplantar NGF-induced hyperalgesia	Significant reduction of hyperalgesia and neutrophil accumulation	[189]
Carrageenan-induced hyperalgesia	Marked time-dependent reduction of mechanical hyperalgesia	[183]
Carrageenan-induced hyperalgesia (s.c. sponge implant)	Significant reduction of new nerve formation and decrease of granuloma-associated hyperalgesia	[184]
Carrageenan-induced hyperalgesia	Significant increased mechanical and thermal thresholds (anti-hyperalgesic effect)	[202]
Formalin-induced nociception	Dose-dependent reduction of nocifensive behaviour in both early and late phases	[202]
Formalin-induced neuropathic-like behaviour	Significant and dose-dependent decrease of mechanical allodynia and thermal hyperalgesia	[185]
Oxaliplatin-induced neuropathic pain	Significant decrease of hyperalgesia and allodynia and improvement in motor coordination	[176]
Streptozotocin-induced diabetic neuropathy	Dose-dependent and significant relief of mechanical allodynia	[186]
Formalin-induced persistent somatic pain	Significant attenuation of the first and early second phases of nociceptive behaviour	[132]
Carrageenan-induced hyperalgesia	Significant reduction of thermal hyperalgesia by 57% (superior effect compared to meloxicam)	[187]
CFA-induced joint pain	Significant decrease of extravasation and mechanical allodynia	[175]
Formalin-evoked persistent somatic pain	Significant attenuation of mechanical allodynia and heat hyperalgesia (over 90%)	[201]
**Visceral Inflammatory Pain**		
Turpentine inflammation of the urinary bladder	Significant attenuation of the vesical hyper-reflexic response	[180]
Acetic acid-evoked writhing	Dose-dependent attenuation of the writhing response	[174]
Turpentine inflammation of the urinary bladder	Dose-dependent attenuation of referred hyperalgesia	[188]
Kaolin-evoked writhing	Potent inhibition of the nocifensive response	[174]
Magnesium sulphate-evoked writhing	Dose-dependent inhibition of the nocifensive response	[174]
NGF-induced inflammation of the urinary bladder	Significant increase of micturition threshold	[182]
PPQ-induced persistent visceral pain	Dose dependent inhibition of visceral pain measured as stretching movement inhibition	[190]
Cyclophosphamide-induced cystitis	Significant decrease of the pain score	[191]

Abbreviations. CFA, Complete Freund’s adjuvant; MIA, monosodium iodoacetate; NGF, nerve growth factor; OA, osteoarthritis; PPQ, phenyl-p-quinone.

**Table 4 animals-11-00952-t004:** Pain relieving effect of PEA—mainly given via intraperitoneal route—in animal models of neuropathic and mixed pain. Summary of studies in chronological order.

Animal Model	Main Behavioural Effect	Ref.
**Neuropathic Pain**		
Partial sciatic nerve injury	Reduction of hyperalgesia (−79.4%)	[192]
Spinal cord injury	Significant reduction of the severity of spinal cord trauma	[193]
Chronic constriction injury	Significant relief of thermal hyperalgesia and mechanical allodynia	[194]
Chronic constriction injury	Significant and time-dependent relief of thermal hyperalgesia and mechanical allodynia (already after two administrations)	[120]
Partial sciatic nerve injury	Restored thermal and mechanical thresholds. Decrease of pain-induced memory deficits	[195]
Diabetic neuropathic pain	Significant antinociceptive effect. Significantly increased thresholds to mechanical stimuli	[196]
Sciatic nerve injury	Reduced nerve edema and inflammatory infiltrate (sub-optimal doses of PEA combined with acetaminophen)	[197]
Partial sciatic nerve injury	Restored cognitive behaviour and reduced cognitive decline associated with neuropathic pain	[198]
Chronic constriction injury	Strong dose-dependent suppression of mechanical allodynia and heat hyperalgesia upon single and repeated (7 consecutive days) administration	[201]
**Chronic mixed pain**		
MIA-induced OA pain	Significant decrease of mechanical allodynia and improved locomotor function	[187]
MIA-induced OA pain	Significantly restored paw withdrawal threshold and weight-bearing compared to the vehicle-treated controls in a dose-dependent fashion	[199]
Vitamin D deficiency-induced chronic pain	Significant reduction of allodynia and neuronal sensitization	[200]

Abbreviations. MIA, monosodium iodoacetate; OA, osteoarthritis.

## 9. Clinical Evidence for Micro-PEA Dietary Supplementation in Pain Relief

On the clinical side, micro-PEA (i.e., micronized, ultramicronized or co-micronized PEA) has been orally administered as a dietary food for special medical purposes to human patients, either singly [206,207,208,209,210,211,212,213,214,215,216,217,218,219,220,221,222,223,224,225] or in combination with (i.e., add-on dietary intervention to) opioids, gabapentenoids or NSAIDs [226,227,228,229,230,231,232,233,234,235,236,237,238,239,240,241,242,243,244,245,246,247,248], as well as antioxidant compounds (e.g., luteolin, quercetin, polydatin) [249,250,251,252,253,254,255,256,257,258,259,260,261] in several painful conditions [53,262].

Altogether, nearly 5000 patients have been clinically investigated so far in dozens of published trials, showing an important overall effect in chronic pain, either neuropathic (Table 5), mixed (Table 6) or pelvic pain (Table 7).

One of the most interesting findings comes from the neurophysiological assessment of 20 patients with chemotherapy-induced painful neuropathy, with daily administered micro-PEA 300 mg/bid for two months. Besides significant pain reduction, increased conduction velocity of myelinated fibers was recorded, with sensory nerve action potentials from sural and ulnar nerves, compound motor action potentials from peroneal and ulnar nerves and laser-evoked potentials for Aδ fibers being significantly improved [212].

A further striking finding comes from the so-called “number needed to treat” (NNT), i.e., a measure depicting the effectiveness of an intervention (the lower the NNT, the more effective the intervention). The calculation was elegantly made by researchers from the Department of Human Neurosciences, “Sapienza” University of Rome [216]. In particular, the percentage of patients who manifested at least 50% pain relief in response to a daily supplementation of micro-PEA 600 mg/die was calculated based on data from a multicenter, double-blind, placebo-controlled, randomized study on 636 patients with low back pain. NNT was found to be 1.7 [216]. It must be pointed out that it is a remarkable NNT value within the broad panorama of treatments for low back pain in human patients. A systematic review on first-line treatments for neuropathic pain has indeed shown that NNT for 50% pain relief ranges from around 4 to 10 across most positive trials (Table 8) [263]. The much lower NNT for micro-PEA (i.e., 1.7) emphasizes the good outcome for neuropathic pain relief. The relevance of the data is further strengthened by the non-significant (and indeed infinite) number needed to harm [216], that is, how many patients must receive a particular treatment for one additional patient to experience a particular adverse outcome.

Overall, micro-PEA has shown a very favorable treatment profile in the management of chronic pain in human patients.

As far as privately owned animals are concerned, two trials have recently dealt with micro-PEA dietary administration for pain relief. The first is a case series in four jumping horses orally supplemented with PEA-um for non-responsive lameness and significant impairment of athletic performance [264]. In particular, the diagnoses were the following: navicular syndrome of the left forelimb (1 case), complicated case of chronic navicular syndrome and OA of the distal interphalangeal joint of the right forelimb (1 case), and OA of the distal intertarsal joint of the right hindlimb (2 cases). Horses were fed daily with PEA-um (2.5 g) mixed with a regular mixture of cereals for four months. At the end of the first month of supplementation, lameness on the AAEP scale (American Association of Equine Practitioners 0–5 scale, with zero indicating no perceptible lameness, and five being most extreme) showed improvement in all horses. Three months later, lameness was graded zero, allowing successful return to showjumping without disease recurrence [264].

The second study is an open-field trial performed in 13 medium-to-large-breed client-owned adult dogs, with chronic OA and persistent lameness longer than one month. All dogs were supplemented for 4 weeks with a complementary feed containing PEA co-ultramicronized with the natural antioxidant quercetin (i.e., PEA-q, 24 mg/kg body weight). The Canine Brief Pain Inventory (CBPI) questionnaire was used to assess the severity of chronic pain (PSS, Pain Severity Score) and how it interfered with the dog’s normal functioning (PIS, Pain Interference Score). With success defined as a reduction of ≥1 in PSS and PIS, treatment was classified as successful in 54.5% dogs as early as week 2 and CBPI scores significantly decreased throughout the study (Figure 7). Moreover, lameness (either scored by the veterinarian on a 0–4 clinical scale or objectively assessed through a dynamic gait analysis) was found to significantly improve during the treatment period [265].

The findings of the trials summarized above provide clinical evidence on PEA-um (eventually co-micronized with quercetin) as a promising treatment option for chronic pain and related functional disability in horses, as well as dogs.

## 10. Conclusions

The management of chronic pain is the burden of veterinary practitioners. Multiple pharmacological agents have been employed to treat diverse pathological pain states, including opiates, NSAIDs, anticonvulsants, antidepressants, and others [29]. However, adverse effects could limit dosing and therapeutic efficacy [163,164].

The recent understanding of the role of non-neuronal cells in pain processing is uncovering potential new targets for managing chronic pain [104]. Furthermore, it is becoming increasingly clear that enhancing endocannabinoid signalling may prevent patients from developing persistent or chronic pain states mainly through non-neuronal cell modulation [266,267,268,269]. One such strategy is the dietetic use of the endocannabinoid-like PEA in bioavailable formulations (i.e., micro-PEA). As reviewed here, there is now strong evidence supporting the dietary supplementation with micro-PEA (either as alternative or add-on to conventional treatment) in the management of chronic pain. Such a critical mass of data is being generated that PEA is currently listed among the novel nonopioid interventions to chronic pain [270].

Although clinical studies in veterinary patients are warranted, the reviewed findings lay the foundations for a scientific and rational use of micro-PEA in the dietary management of chronic pain in dogs and cats.

## Figures and Tables

**Figure 1 animals-11-00952-f001:**
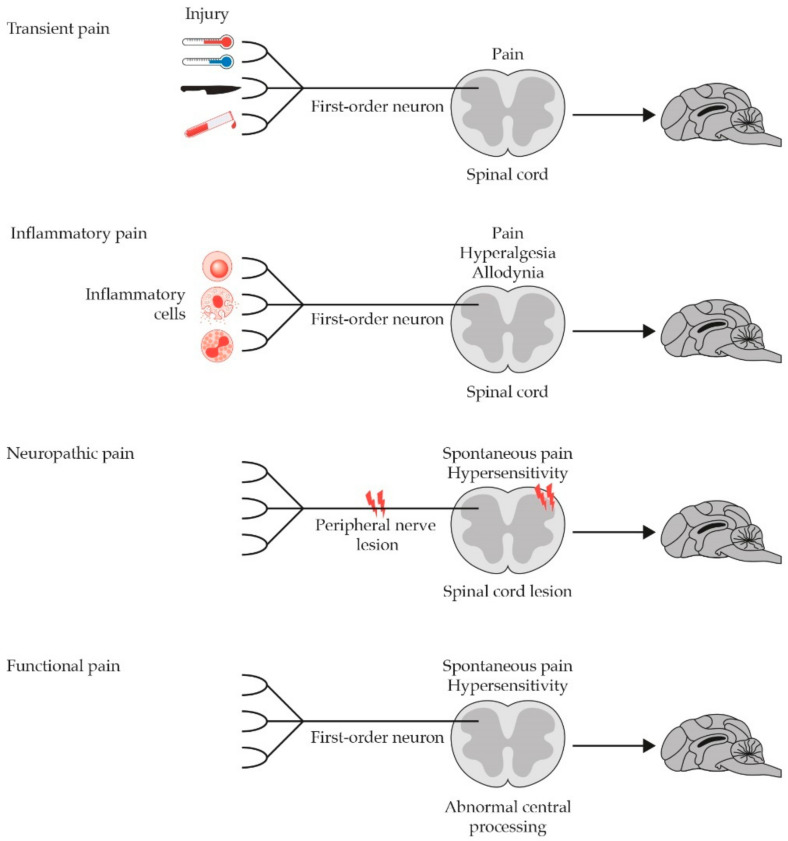
Schematic representation of the four different types of pain, based on their etiopathogenesis. Modified from [29].

**Figure 2 animals-11-00952-f002:**
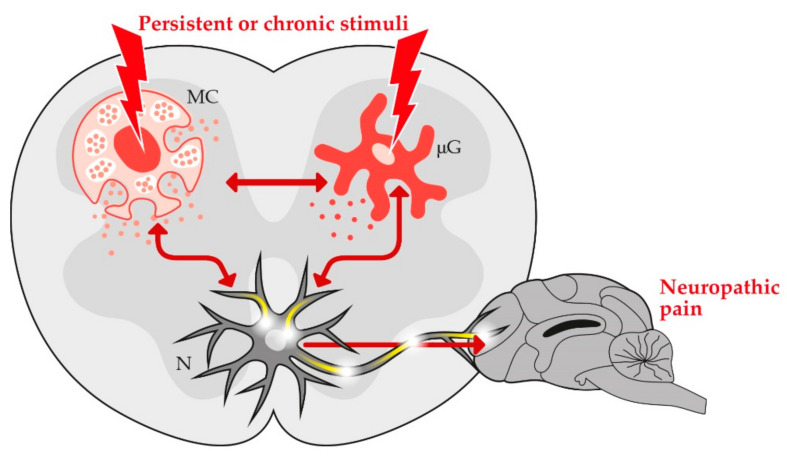
Once hyper-activated spinal microglia and mast cells release a wide variety of mediators able to induce chronic neuronal hypersensitivity (i.e., central sensitization) and the resulting neuropathic pain. MC, mast cell; μG, microglia; N, neuron.

**Figure 3 animals-11-00952-f003:**
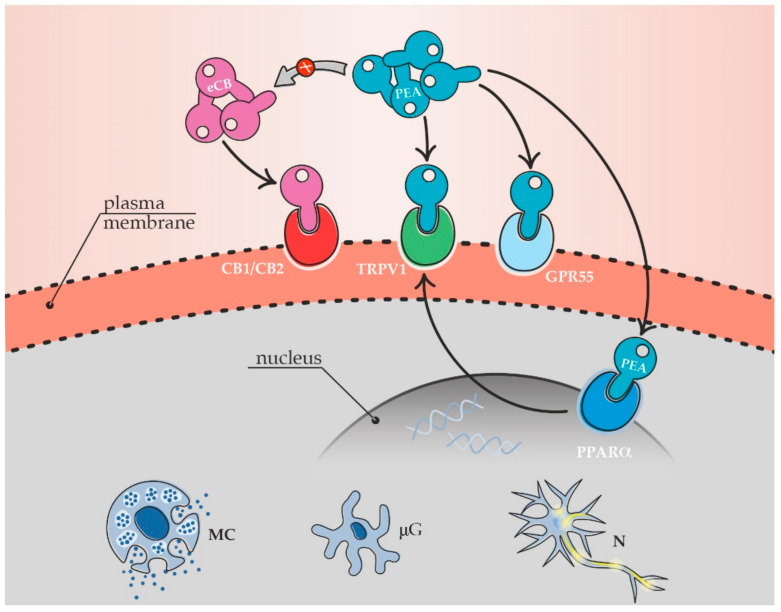
Direct and indirect agonism of PEA (blu key) on canonical (CB1, CB2) and putative (TRPV1, GPR55, PPARα) cannabinoid receptors expressed on the plasma membrane and/or nucleus of neuronal and non-neuronal cells. The multitarget receptor mechanism allows for the physiological control of pain pathways by PEA. (eCB, endocannabinoids, e.g., anandamide, AEA and 2-arachydonoylglycerol, 2-AG; MC, mast cell; μG, microglia; N, neuron).

**Figure 4 animals-11-00952-f004:**
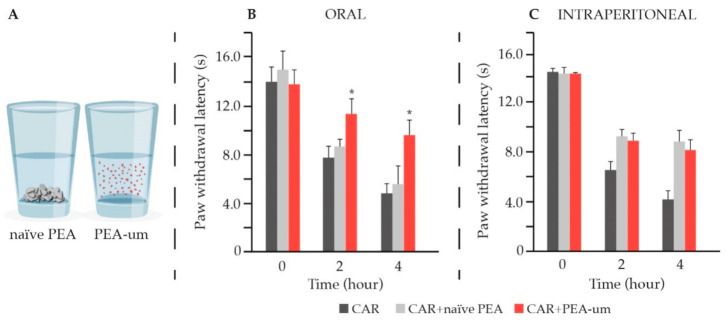
Advantages of PEA micronization. Reducing particle size increases particle surface area, resulting in higher dissolution rate of micronized PEA compared to the naïve form (**A**). In the carrageenan-induced hyperalgesia (CAR) PEA-um exerted a superior anti-hyperalgesic effect compared to naïve PEA after oral administration (**B**). On the contrary, no difference was observed after intraperitoneal administration (**C**). * *p* < 0.01 vs. CAR. Modified from [172].

**Figure 5 animals-11-00952-f005:**
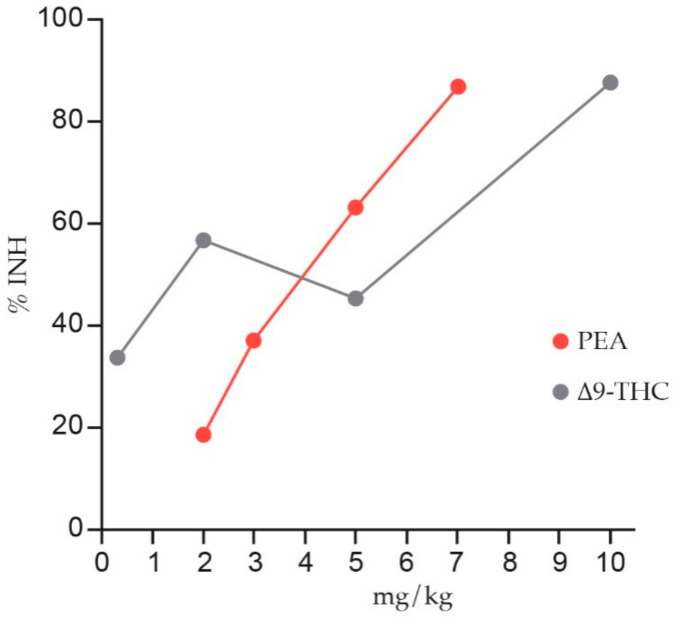
Anti-nociception elicited by Δ9-THC and PEA after intraperitoneal administration in a visceral pain model (phenyl-p-quinone, PPQ). The dose response curves for percentage inhibition of stretching movements (%INH) are reported. Δ9-THC and PEA were administered 15 min and 10 min prior to PPQ, respectively. Redrawn from [190].

**Figure 6 animals-11-00952-f006:**
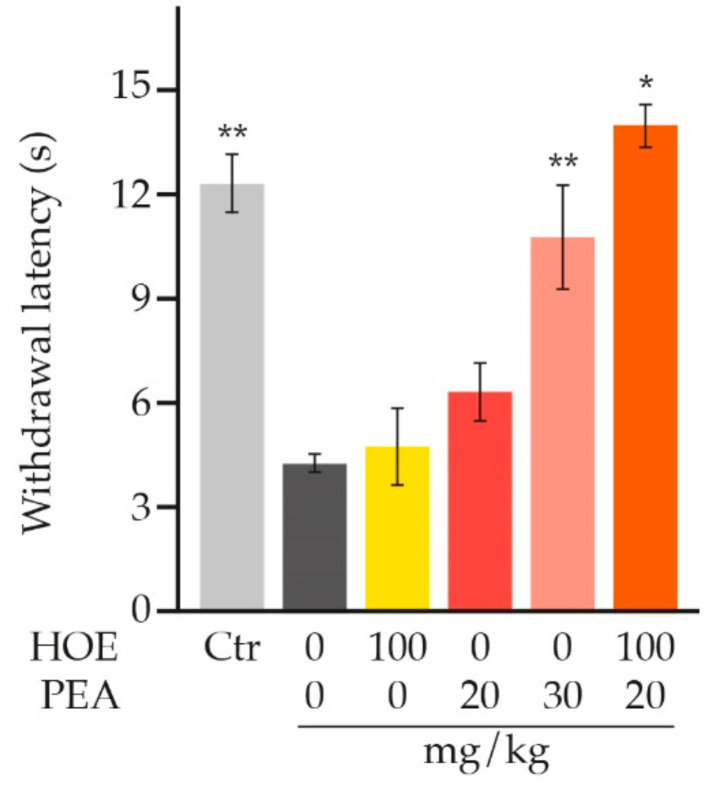
Effects of combining a single sub-optimal oral dose of hemp oil extract (HOE) with PEA on heat hyperalgesia associated with neuropathic pain. Oral administration of PEA (30 mg/kg) significantly relieves heat hyperalgesia, increasing the withdrawal latency to nearly control values (Ctr), while HOE (100 mg/kg) does not exert any effect. The combination of the two compounds at the indicated doses (orange bar) exerts greater-than-additive antinociceptive effects. * *p* < 0.001 and ** *p* < 0.0001 vs. chronic constriction injury (dark grey bar). The source data come from Figure 2B, Figure 4D and Figure 6B published in [201].

**Figure 7 animals-11-00952-f007:**
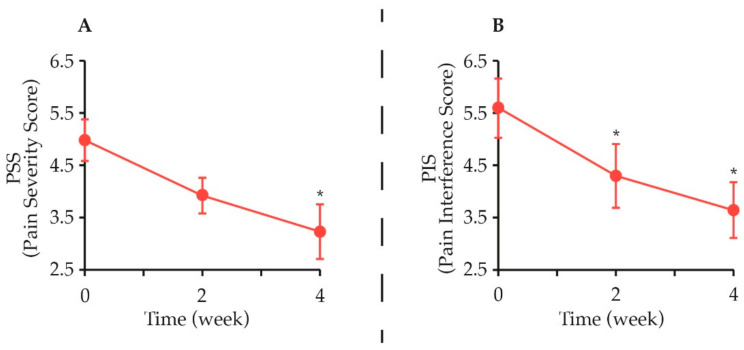
Dietary administration of PEA-q to privately owned dogs with chronic pain reduced the CBPI score. (**A**) During the four-week treatment, the mean severity of pain on PSS decreased significantly (*, *p* = 0.023). (**B**) The decrease of mean PIS was already statistically significant at the first control (week 2) and maintained a statistically significant decrease at the end of the study (week 4) (*, *p* = 0.009 for both comparisons). Drawn from data presented in [265].

**Table 1 animals-11-00952-t001:** Mast cell and microglia ID chart.

	Mast Cells	Microglia
Cell type	Resident long-lived immune-inflammatory cells [57,58]	Resident long-lived immune-inflammatory cells [59,60]
Location	Periphery In association with sensory nerves, forming synapse-like structures, in virtually any tissue, especially those exposed to the environment PNS Within nerves (endoneural mast cells) [61,62,63,64]	CNS Throughout the brain and spinal cord (largely outnumbering neurons), where they provide nourishment to neurons, regulate neural activity and generate innate immune responses [65]
	CNS Spinal meninges; different brain parenchymal sites (e.g., hippocampus and thalamic, hypothalamic region); blood brain barrier (brain side), generally located near microglia [66,67,68]	
Activation kinetics	Rapid release of prestored mediator in response to stimuli (e.g., sensory nerve activation), thanks to a wide range of receptors Release more than 50 mediators with vasoactive, neurosensitizing and pro-inflammatory effects [69,70,71,72]	Become activated in response to local stress (e.g., nerve injury), shifting their phenotype from a quiescent to an activated state Release pro-inflammatory cytokines and chemokines in the brain and spinal cord [73,74]
Type of pain involved in	Inflammatory and neuropathic pain, either visceral and somatic, e.g., osteoarthritis pain, discogenic pain, viscerovisceral hyperalgesia [75,76,77,78,79,80,81,82,83,84,85,86,87,88]	Neuropathic pain (e.g., canine intervertebral disk disease); also involved in allergic-induced neuropathic pain, acute inflammatory pain, paradoxical pain associated with long-term opioid administration [59,89,90,91,92,93]

Abbreviations. CNS, central nervous system; PNS, peripheral nervous system.

**Table 2 animals-11-00952-t002:** Main causes of maladaptive pain in dogs and cats. From [148].

**Main Causes of Inflammatory Pain**
Chronic lesions/inflammations affecting superficial tissues (skin, mucous membranes, teeth, some portions of the eye) and deep somatic tissues (bones, muscles, joints)
Chronic ulcers at skin, mucous, or corneal sites
Chronic inflammatory diseases
Gingivostomatitis Periodontitis Pulpits Otitis Conjunctivitis Keratitis Osteoarthritis
Myofascial trigger points
Discs herniation
Somatic cancers (skin, breast, osteosarcoma)
Chronic injury/inflammation affecting deep visceral tissues
Chronic inflammatory diseases Inflammatory bowel disease (IBD) Pancreatitis Cystitis (i.e., feline idiopathic cystitis) Prostatitis
Gastrointestinal ulcers
Cancers affecting visceral districts Primary visceral cancer Metastatic invasion of viscera
**Main Causes of Neuropathic Pain**
Peripheral and central nervous system disorders
Poliradiculoneuritis
Diabetic neuropathy
Disk compression radiculopathy with nerve damage
Tumor infiltration neuropathy
Paraneoplastic neuropathies
Myelin sheath cancer
Central nervous system (CNS) cancers
Chronic visceral pathologies with neuropathic component
Chronic pancreatitis
IBD
Feline interstitial cystitis
Visceral cancers

**Table 5 animals-11-00952-t005:** Pain relieving effect of micro-PEA (i.e., PEA-m or PEA-um) on chronic neuropathic pain: overview of human trials in chronological order.

Diagnosis (Trial Design)	No. of Pts	Dose	Main Result	Ref.
**Peripheral neuropathic pain**				
Sciatic pain due to radicular and/or core compression of the sciatic nerve and discopathy (Double-blind, randomized, two doses of micro-PEA vs. placebo)	636	300 mg/die or 300 mg/bid for three weeks	Significant decrease of pain on VAS (from 7 to 2)	[206]
Diabetic neuropathy pain associated with carpal tunnel syndrome (Group-controlled, randomized, micro-PEA treatment vs. standard care)	50	600 mg/bid for two months	Significant relief of pain. Significant improvement of neurophysiologic parameters	[207]
Painful neuropathies (Open-label study)	27	300 mg/bid for three weeks, followed by 300 mg/die for four weeks	Significant reduction of pain and improvement of electrophysiological parameters	[208]
Sciatic pain (Double-blind, randomized, two doses of micro-PEA vs. placebo (as an add-on therapy))	111	300 mg/die or 300 mg/bid for three weeks	Significant decrease in pain severity and duration of treatment with anti-inflammatory and analgesic drugs	[226]
Neuropathic chronic pain (diabetic neuropathy and postherpetic neuralgia) (Open, combination therapy with GBPs)	30	600 mg/bid for 45 days	Significant decrease of pain on VAS (from 7.6 to 1.8)	[227]
Low back pain (Group-controlled (add-on therapy to standard analgesics))	81	600 mg/bid for three weeks followed by 600 mg/die for four weeks	Significant reduction of pain intensity compared to control group	[228]
Sciatic pain (Group-controlled, randomized, add-on therapy to standard analgesics)	85	300 mg/bid for 30 days	Significant relief of pain (scored both on VAS and Oswestry Low Back Pain Scale) compared to the analgesic-only group	[229]
Diabetic neuropathic pain (Open-label study)	30	300 mg/bid for two months	Significant reduction of pain, burning, paraesthesia and numbness	[209]
Carpal tunnel syndrome in diabetic pts (Group-controlled, randomized vs. non-treated pts)	40	600 mg/bid for two months	Significant reduction of pain and improvement of functional status and neurophysiologic parameters	[210]
Pain associated with carpal tunnel syndrome (Group-controlled, randomized, two doses of micro-PEA vs. non-treated pts)	26	1st arm: 300 mg/bid for 30 days 2nd arm: 600 mg/bid for 30 days	Significant dose-dependent reduction of pain and improvement of neurophysiologic parameters compared with control group	[211]
Chemotherapy-induced painful neuropathy (Open-label study)	20	300 mg/bid for two months	Significant pain reduction on NRS and significantly increased conduction velocity of myelinated fibers on neurophysiological assessment	[212]
Low back pain (Open, combination therapywith OPI)	20	600 mg/bid for 30 days	Significant decrease of pain on VAS (from 7 to 2.5)	[230]
Various chronic pain-associated disorders (Open, combination therapy with GBPs and OPI)	517	600 mg/bid for three weeks followed by 600 mg/die for four weeks	61% decrease of mean pain score on NRS	[231]
Diabetic neuropathic pain (Group-controlled (micro-PEA + GBPs vs. GBPs))	74	600 mg/bid for the first 10 days, then 600 mg/die for 20 days, followed by 300 mg/die for 30 days	Significantly higher rate of responders (i.e., >60% decrease in pain score) compared to GBP group	[232]
Chronic neuropathic pain from lumbosciatica (Multicentral prospective, group-controlled study (add-on therapy to standard analgesics vs. standard analgesics))	118	300 mg/bid for 30 days	Significantly larger improvements in VAS and QoL compared to standard therapy alone	[233]
Chronic pain associated to different pathological conditions (Observational study (add-on to poorly effective standard analgesics))	610	600 mg/bid for three weeks + 600 mg/die for the following four weeks	Significant decrease of the mean score of pain on NRS (even in pts without concomitant analgesics)	[234]
Diabetic neuropathy (Open-label study)	30	300 mg/bid for two months	Significant decrease of pain severity and related symptoms evaluated by Michigan Neuropathy Screening instrument and NPSI	[213]
Diabetic or traumatic chronic neuropathic pain, with VAS greater than 6 in spite of the best therapeutic regimen with GBPs and/or OPI (Open-label study (add-on))	30	1200 mg/die for 40 days	Significant and time dependent decrease of pain on VAS and NPSI, as well as QoL on EQ-5D	[235]
Pain associated to fibromyalgia syndrome (Retrospective + prospective study (SNRI + GBPs vs. SNRI + GBPs + micro-PEA))	80	600 mg/bid in the first month and 300 mg/bid in the next two months	Further reduction in the number of positive tender points and significant reduction in pain, compared to SNRI + GBPs only	[236]
Low back pain (Case report (combined to low dose SNRI))	2	600–1200 mg/bid for two months	Significant decrease of pain on NRS	[237]
Failed back surgery syndrome (caused by laminectomy, discectomy, or vertebral stabilization) (Observational study (add-on to 1-month standard analgesic treatment, i.e., OPI + GBPs))	35	1200 mg/die for the first month and 600 mg/die for the second month	Further and significant decrease in pain intensity compared to the first month of standard analgesics	[238]
Chronic, non-cancer, non-ischemic pain in the back, joints or limbs in elderly pts (≥ 65 years) (Series of N-of-1 randomized trials)	10	600 mg/bid	Statistically significant favorable impact on either pain intensity or function impairment in some of the three of the pts	[239]
Chronic low back pain (Two arm (prospective and retrospective), pilot observational study (add-on to OPI compared to OPI only))	55	600 mg/bid for six months	Significantly higher reduction in: - pain intensity on VAS - neuropathic component (on DN4 questionnaire) - degree of disability (on Oswestry Disability Index) - OPI dosage assumption compared to OPI only group	[240]
Neuropathic pain associated with nonsurgical lumbar radiculopathies (Retrospective study (add-on to 4-day treatment with ACT + OPI))	100	600 mg/bid for 30 days followed by 600 mg/die for 30 days	Significant pain relief in pts with mild, moderate and severe baseline painful symptoms	[241]
Neuropathic pain associated with nonsurgical lumbar radiculopathies with X-ray signs of spondylosis and CT/MRI signs of IVD protrusion or dehydration (Prospective single-blind (add-on to 7-day treatment with fixed combination ACT + OPI))	155	1200 mg/die for 30 days. If unsuccessful, further 30 days with 600 mg/die followed, if needed, by a second cycle of ACT + OPI for 30 days	Significant improvement of pain and disability after 30 or 60 days depending on the baseline pain severity (VAS 3–8). In pts with baseline VAS ≥9 the second ACT + OPI cycle was needed.	[242]
Carpal tunnel syndrome (Open, controlled study (PEA-um + surgery vs. surgery only))	42	600 mg/bid for 2 months before and 2 months after surgery + 600 mg/die for 30 days	Significant improvement in painful symptoms and overall sleep quality on PSQI	[243]
Burning mouth syndrome (Case report (add-on to poorly effective GBPs))	1	600 mg/bid for three months	Significant decrease of pain on VAS (from 9 to 5). Great reduction of the frequency of episodes	[244]
Chronic orofacial neuropathic pain (post-traumatic neuropathy) (Open-label clinical trial)	22	300 mg/tidfor six weeks	Overall reduction in ongoing pain on VAS. Normalized activity patterns in the ascending pain pathway	[214]
Burning mouth syndrome (Preliminary randomized double-blind controlled trial)	35	600 mg/bid for two months	Statistically significant higher reduction of burning mouth sensation on NRS compared to placebo	[215]
Fibromyalgia Syndrome (Retrospective observational study (add-on to concomitant pharmacological therapy, i.e., SSRI (n = 71), SNRI (n = 66), GBPs (n = 41), TCA (n = 40), BZD (n = 94),OPI (n = 78), NSAIDs (n = 87), MR (n = 35), ACT (n = 45))	407	600 mg/tid for 10 days followed by 600 mg/bid for 20 days followed by 600 mg/die for 125 months	Statistically significant decrease of pain on VAS and statistically significant improved QoL on FIQ	[246]
Low back pain—sciatica (Post-hoc analysis of a placebo-controlled study)	600	600 mg/die	NNT of 1.7 (1.4–2) for the effect on pain and 1.5 (1.4–1.7) for the effect on function	[216]
Chronic low back pain (i.e., lumbo-sciatica and lumbo-cruralgia due to multiple herniated discs in the lumbar spine) (Open, add-on to standard analgesics + functional rehabilitation session)	120	600 mg/bid for 20 days, followed by 600 mg/die for 40 days	Significant decrease of pain intensity scores (from 6.3 ± 0.1 at baseline to 3.7 ± 0.09 and 2 ± 0.09 at 30 and 60 days, respectively)	[245]
**Central neuropathic pain**				
Neuropathic pain associated with multiple sclerosis (Open-label study)	20	300 mg/bid for two months	Significant decrease of neuropathic pain	[217]
Neuropathic pain and spasticity in post-stroke pts (Open, controlled micro-PEA + P vs. PT only)	20	600 mg/bid for two months followed by 600 mg/die for 30 days	Significant decrease of pain and spasticity	[247]
Pain associated with stroke (Observational study (co-um PEA-Lut in association with the stroke therapy, e.g., thrombolytics))	250	700 mg + 70 mg for two months	Pain on NRS halved after 30 days	[249]
Migraine without aura—at least 6 months’ duration (Open-label study)	50	600 mg/bid for three months	Significant decrease of - day per month with migraine - pain intensity - amount of analgesics; Reduction of hypothermia and response to trigger factors (thermography)	[218]
Nummular headache (Case report (add-on to decreasing topiramate dose))	1	600 mg/die	Improvement in pain symptoms and on pain measuring scales	[219]
Occipital Neuralgia (Case report])	1	1200 mg/die	Significant improvement of pain, after around 2 weeks of treatment	[220]
Migraine with Aura (Single blind study (add-on to acute NSAIDs, i.e., ibuprofen, diclofenac sodium, or nimesuilde for about 2 days during acute migraine attack))	20	1200 mg/die for three months	Statistically significant and time-dependent pain relief, already evident at 60 days and lasting until the end of the study	[221]
Migraine without aura in a pediatric population (Open-label pilot study)	70	600 mg/die for three months	Significant decrease of - the number of monthly attacks - the mean intensity of attacks - percent of pts with severe attacks - monthly assumption of drugs for the attacks	[222]

Abbreviations. ACT, Acetaminophen; bid, *bis in die* = twice daily; BZD, benzodiazepines; co-um PEA-Lut, co-ultramicronized palmitoylethanolamide and luteolin; CT, computed tomography scans; die, daily; EQ-5D, Health Questionnaire Five Dimensions; FIQ, Fibromyalgia Impact Questionnaire on quality of life; GBPs, gabapentinoids; IVD, intervertebral disk; MR = muscle relaxants; MRI, magnetic resonance imaging; NNT, Number Needed to Treat; NPSI, Neuropathic Pain Symptom Inventory; NRS, Numeric Rating Scale; NSAIDs, non-steroidal anti-inflammatory drugs; OPI, opiates; PSQI, Pittsburgh Sleep Quality Index; PT, physiotherapy; pts, patients; QoL, quality of life; tid, ter in die = three times daily; SSRI, serotonin selective reuptake inhibitors; SNRI, serotonin noradrenaline selective inhibitors; TCA, tricyclic antidepressants; VAS, visual analogue pain scale.

**Table 6 animals-11-00952-t006:** Pain relieving effect of micro-PEA (i.e., PEA-m or PEA-um) on chronic mixed pain: overview of clinical trials in chronological order.

Diagnosis [Trial Design]	No. of Pts	Dose	Main Result	Ref.
TMJ pain caused by OA (Double-blind randomized vs. NSAIDs)	24	300 mg in the morning + 600 mg in the eveningfor 7 days; followed by 300 mg/bid for 7 days	Significant decrease of pain on VAS (from 7 to 0.7) and significantly improved maximum mouth opening compared to NSAIDs	[223]
OA-induced TMJ arthralgia (Case series (initially combined with NSAIDs))	12	600 mg/die (together with NSAIDs for the first 4 days, then singly for the following 10 days)	Significant pain reduction after 4 days. Significant improvement of maximum mouth opening	[248]
Knee OA pain (Double-blind randomized placebo-controlled study (two doses))	111	300 mg/die or 600 mg/die	Significant reduction of WOMAC, pain on NRS and anxiety	[224]
Pain in arthrogenic TMJ dysfunction and similar disorders (Systematic review of 5 studies (4 RCTs + 1 retrospective cohort study))	227	300 mg/die and over	Effective in arthrogenic TMJ dysfunction and related disorders, with a superior analgesic effect to some NSAIDs and a low rate of adverse events	[225]

Abbreviations. bid, *bis in die* = twice daily; die, daily; NSAIDs, non-steroidal anti-inflammatory drugs; pts, patients; OA, osteoarthritis; RCTs, randomized clinical trials; TMJ, temporomandibular joint; VAS, visual analogue pain scale; WOMAC, Western Ontario and McMaster Universities Osteoarthritis Index.

**Table 7 animals-11-00952-t007:** Pain relieving effect of PEA-Pol (i.e., PEA co-micronized with the antioxidant polydatin in 10:1 ratio) on chronic pelvic pain: overview of clinical trials in chronological order.

Diagnosis [Trial Design]	No. of Pts	Dose	Main Result	Ref.
Chronic pelvic pain associated with endometriosis/dysmenorrhea/ interstitial cystitis (Open-label study)	25	(200 + 20) mg/tid for 40 days	Significant reduction of pain on VAS (from 6.8 to 1.7); significant decrease in the use of NSAIDs.	[250]
Adolescent primary dysmenorrhea (Open-label study)	20	(400 + 40) mg/bid for six months	70% decrease in pelvic pain	[251]
Chronic pelvic pain and dyspareunia associated with endometriosis (Open (case series))	4	(200 + 20) mg/bid for three months	Significant decrease of pelvic pain and dyspareunia; significant reduction in the use of analgesics.	[252]
Pudendal neuralgia (Case report)	1	PEA-um 300 mg/tid gradually decreased to 300 mg/die for one year	Resolution of chronic pelvic pain	[253]
Chronic pelvic pain associated with endometriosis (Double-blind, randomized parallel-group (celecoxib), placebo-controlled)	61	(400 + 40) mg/tid for three months	Significant decrease of chronic pelvic pain, dysmenorrhea and dyspareunia	[254]
Endometriosis associated with severe pelvic pain (Open-label study)	24	(400 + 40) mg/bid for three months	Statistically significant decrease of pain, dysmenorrhea and dyspareunia and improved QoL, as well as decreased assumption of NSAIDs	[255]
Pain related to endometriosis (Prospective study)	47	(400 + 40) mg/bid for three months	Significant decrease of chronic pelvic pain, dyspareunia and dysmenorrhea on VAS since the first visit (day 30)	[256]
Vestibulodynia (Randomized, placebo-controlled, combined with TENS)	20	(400 + 40) mg/bid for two months	Significant decrease of pain on VAS in both groups. Superior decrease of current perception threshold for C fibers in treated (40%) compared to placebo group (4.6%)	[257]
Primary dysmenorrhea (Randomized placebo-controlled with follow-up)	220	(400 + 40) mg/die for 10 days (from the 24th day of cycle)	Improvement of pelvic pain in 98% of cases in the treated group vs. 56% in the placebo group. Statistically superior effect compared to placebo	[258]
Irritable bowel syndrome (Randomized double-blind placebo-controlled)	54	(200 + 20) mg/bid for 12 weeks	Reduction of abdominal pain and discomfort	[259]
Symptomatic women with laparoscopic diagnosis of endometriosis (Single-arm, open-label)	30	(400 + 40) mg/bid for 80 days, after 10 days PEA-um 600 mg/bid	Significant decrease of symptoms (pain on VAS, dysmenorrhea, dyspareunia, and dyschezia, dysuria); increased QoL and psychological well-being; significant reduction in the use of the analgesics	[260]
Interstitial cystitis/bladder pain syndrome (IC/BPS) (Pilot, open-label bicentric study)	32	(400 + 40) mg/tid for three months followed by (400 + 40) mg/die for three months	Significant decrease of pelvic pain intensity on VAS from 6.9 ± 0.4 to 4.6 ± 0.4 (the effect persisting up to two months after treatment withdrawal); PUF significantly and progressively decreased; significant reduction in urinary frequency	[261]

Abbreviations. bid, *bis in die* = twice daily; die, daily; NSAIDs, non-steroidal anti-inflammatory drugs; PUF, Pelvic Pain and Urgency/Frequency Symptom Scale; Pts, patients; QoL, quality of life; tid, ter in die = three times daily; TENS, transcutaneous electrical nerve stimulation therapy; VAS, visual analogue pain scale.

**Table 8 animals-11-00952-t008:** NNTs for micro-PEA and the main first-line treatments for neuropathic pain (i.e., the number of patients to treat in order to obtain one patient with at least 50% pain relief) [216,263].

Intervention	NNT
Micro-PEA	1.7
TTAs	3.6
SNRIs	6.4
Gabapentin	6.3
Pregabalin	7.7

Abbreviations. TTAs, tricyclic antidepressants; SNRIs, serotonin-norepinephrine reuptake inhibitors.

## Data Availability

Not applicable. No new data were created or analyzed in this study.

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
