# Peer review of "Chronic Pain in Dogs and Cats: Is There Place for Dietary Intervention with Micro-Palmitoylethanolamide?"

_animals, 2021, doi:10.3390/ani11040952_

Round 1
Reviewer 1 Report
Thank you for a nice review on the important topic of chronic pain.
Some minor comments:
- please remove acupuncture as treatment option unless you have references that proof effectiveness in dogs and cats
- R40-41: "much evidence" is subjective. Please give a grade of evidence according to the EBVM pyramid
- R86-89: please rephrase: "interested veterinarians" is redundant, as they are already reading the paper, what is "valuable information" and it "might help" This makes the statement very broad and vague, and pointless.
- R468-469: same comment about the level of evidence, rather than stating "substantial"
Author Response
Dear Reviewer,
many thanks for your thorough review of our manuscript.
Below you can find a point-by-point response to yours comments.
- Thank you for pointing this out; we did not realize references on the effectiveness of acupuncture were missing. A couple of references on the topic have now been added (please see Reff. 32 and 33).
- Comment on Lines 40-41: reworded as requested (although we don’t consider we can refer to EBVM, given the paucity of currently available studies in veterinary patients).
- Comment on Lines 86-89: the sentence has been rephrased following your suggestion.
- Comment on Lines 468-469: based on your suggestion and the comment of Referee 2, the sentence has been changed.
Reviewer 2 Report
GENERAL COMMENTS:
The manuscript describe the potential role of palmitoylethanolamide (PEA) in controlling pain through non-neuronal cell modulation in animal models, humans and companion animals, laying the foundations for a scientific and rational use in dogs and cats. The topic is of great interest, given the need of veterinarians (and pet parents) to identify synergistic strategies for the treatment of chronic pain in pets. Moreover, it clearifies difference sources and type of pain by providing a graphical summary, easily understandable.
The review is well written, comprehensive and easy-to- read, despite the abundance of brachets in the text (needed to better explain certain definitions and terms). I would suggest the authors, if possible, to limit their use in order to make the reading flowing "smoother".
Sections are well organized and tables/figures adequately structered. I would rather prefer their inclusion at the end of the section in which they are included to not abruptly interrupt the paragraph (unless otherwise indicated by the editorial guidelines)
References cover a wide range of years, from corner-stone publications to recent ones. Literature specifically concerning PEA administration in companion animals is limited, but there is still a lack of knowledge around their efficacy in pets and this review could contribute to improve this gap.
MINOR COMMENTS:
lines 12-15: abundance of brachets make the text difficult to comprehend. Please rewrite in a clearer manner
lines 15-17: currently....currently: please search for a synonym.
line 40: There is much evidence to say there is place. Too colloquial, please rewrite
lines 56-61: citations made too liberally (from 2 to 28 in only eight lines). I would rather prefere a more detailed citation
figure 4: I would rather suggest to delete it from the text because it could be redundant. Insert in the text info provided in figure
table 2: IBD stands for INFLAMMATORI bowel disease not INTESTINAL, please correct
lines 292-292: issue....issue; please find a synonym
line 473: Although clinical studies in veterinary patients are warranted, the reviewed findings lead one to conclude that yes... I consider this sentence too colloquial, please modify
Author Response
Dear Reviewer,
Thank you for your comments and helpful advices, which we believe have improved our paper.
Below you can find a point-by-point response to yours comments.
- Lines 12-15: since tolerance is not a core concept of the review, it has been deleted from the summary as have the relative brackets. Ease-of-reading should be improved now.
- Lines 15-17: a synonym has been used instead of repeating “currently”.
- Line 40: following your comment, the sentence has been rephrased.
- Lines 56-61: references have been moved closer to the relative text.
- Figure 4: according to your suggestion, Figure 4 has been deleted.
- Table 2: IBD has been corrected.
- Line 292: different word for “issue” has been used.
- Line 473: the sentence has been rephrased.
Reviewer 3 Report
Acylethanolamides are lipid substances widely distributed in the body, generated from a membrane phospholipid precursor, N-acylphosphatidyl-ethanolamine (NAPE). The identification of arachidonoyl ethanolamide (anandamide) as an endogenous cannabinoid agonist ligand has focused attention on acylethanolamides. The identification of related additional acylethanolamides with signaling function, such as oleoylethanolamide (OEA) and palmitoylethanolamide (PEA) launched further intensive research. Most of the biological functions of anandamide are mediated by the two G protein-coupled cannabinoid receptors identified to date, CB(1) and CB(2), with the transient receptor potential vanilloid-1 receptor being an additional potential target. There has been increasing pharmacological evidence for the existence of additional cannabinoid receptors, with the orphan G protein-coupled receptor GPR55 and GPR119. The latter receptor can recognize PEA, but altough is structurally related to anandamide, do not interact directly with classical cannabinoid receptors. Instead, it has high affinity for the nuclear receptor PPARalpha, which is believed to mediate many of its biological effects.
The authors in this systematic review paper presented in detail the applications of PEA, especially micronized PEA, in veterinary practice for dogs and cats. The manuscript is well structured, clear and followable for the reader. The figures and tables adequately help the reader to navigate between the data. The article also illustrates well that the effective treatment of different types of pain is a complex task for our pets, dogs and cats, and a challenge for the veterinarian. In my opinion, this summary work fits well with the aims of the journal, so I recommend the acceptance of the article.
My suggestion to improve further the quality of the manuscript:
--A very fresh paper closely related to the topic should be included in the reference list: Int. J. Mol. Sci. 2021 Jan 21; 22(3):1034. doi: 10.3390/ijms22031034. Dong-Soon Im:
GPR119 and GPR55 as Receptors for Fatty Acid Ethanolamides, Oleoylethanolamide and Palmitoylethanolamide
--Results confirming the presence of TRPV1, GPR55, GPR119 and PPARalpha receptors in both species studied here should be presented in text or short tabular form.
Author Response
Dear reviewer, many thanks for your thorough review.
Below you can find a point-by-point response to your comments.
- Thank you very much for the suggested reference – which indeed was originally included at ref. 128 (now ref . 130)
- Following your suggestion, a brief paragraph on the localization of cannabinoid receptors in dogs and cats has been added (page 6, before figure 3) and the relative references are now included (please see references 133-143).